# Purchasing Counterfeits and Citizenship: Public Service Motivation Matters

**Kwangho Jung** [1,*] **, Seung-Hee Lee** [2] **and Jane Workman** [2]

[1]  Korea Institute of Public Affairs, Institute of Information Knowledge and Policy, Graduate School of Public Administration, Seoul National University, 1 Gwanak-ro, Gwanak-gu, Seoul 08826, Korea

[2]  Fashion Design and Merchandising, Southern Illinois University, 311 Quigley Hall, Carbondale, IL 62901, USA; shlee@siu.edu (S.-H.L.); jworkman@siu.edu (J.W.)

*  Correspondence: kwjung77@gmail.com or kwjung77@snu.ac.kr; Tel.: +82-02-880-5626

**Abstract:** The purpose of this study was to examine how consumers' public service motivation (PSM) is related to ethical consumption behaviors and how past experience of unethical behavior can reduce the impact of PSM on ethical consumer behaviors. A nationally representative sample from South Korea was used to explore how PSM influences willingness to purchase fashion counterfeits and how the impact of PSM differs for those with and without past experience buying fashion counterfeits. Higher PSM was associated with less willingness to buy counterfeits. Past experience buying counterfeits was associated with greater willingness to buy counterfeits. Past experience buying counterfeits intervened between the impact of PSM and willingness to buy counterfeits such that the impact of PSM was weakened.

**Keywords:** consumer ethics; public service motivation; civic virtue; experience buying counterfeits; moral costs

## 1. Introduction

The history of business ethics research has illustrated two main emphases: supplier ethics and consumer ethics. While corporate social responsibility has received much attention, systematic research on demand side ethics from the consumers' perspective is still limited. Ethics studies regarding consumers have incorporated two different approaches: positive consumption derived from ethical business processes and products (i.e., ethical consumerism) [1] and negative consumer behaviors such as deception, changing price tags, insurance fraud, and intellectual property theft (i.e., consumer misbehavior) [2,3]. However, global markets provide ethical issues for both suppliers and consumers. Research on consumer misbehavior is compatible with the economic theory that if there is little demand for illegal goods, the supply will dry up. Research on ethical consumer behavior can provide policy implications for the elimination or reduction of businesses producing or selling illegal goods.

Despite numerous attempts to educate consumers about consumer ethics, the counterfeit market continues to grow. Trade in counterfeit goods has increased from $250 billion annually in 2008 to more than $461 billion in 2013; an increase of over 80% in a five-year period [4]. An important research question is: How do consumer ethics affect consumers' willingness to buy counterfeits and what particular aspects of consumer ethics can reduce willingness to buy counterfeits?

Recent research on consumer ethics has addressed how ethical factors reduce or control the purchase of counterfeits and consumer fraud [5–7]. Scales measuring unethical consumption behaviors have been developed to identify ethical components that are more or less important, either directly or indirectly. For instance, the Muncy-Vitell consumer ethics scale [2] measures the extent to which consumers believe certain questionable situations are ethical or unethical. In addition, consumer ethical

ideologies such as idealism and relativism have been explored regarding relationships with counterfeit purchase [8]. However, none of these scales directly measure what kind of ethical motivations are associated with the decision making process of purchasing counterfeits.

Indirect measures of consumer ethics lead to a lack of theoretical usefulness and coherence. Previous studies have used consumer ethical perceptions, for example, perceptions regarding the degree to which certain questionable behaviors are ethical/unethical [2]. More recent studies have considered multiple dimensions of consumer ethics including moral judgment, moral intensity, and moral affect for purchasing counterfeits [9]. Recent ethical scales include items measuring the degree of willingness to devote extra time and cost for ethical consumption as well as the desire to be an ethical consumer. However, much consumer behavior is self-centered with little thought given to far-reaching consequences of purchasing decisions such as the effects of counterfeits on a country's economy. The current study has three distinct aspects compared to previous research on consumer ethics and counterfeits.

First, the public service motivation (PSM) concept is reviewed and the PSM scale is tested as to whether it is related to willingness to buy counterfeits via a strong service motivation for public interest. The concept of PSM consists of several sub-dimensions representing civic virtue including attraction to public policy making, public interest, compassion, and social justice. Research on a relationship between PSM and consumption behaviors provides a new potential link on how civic engagement can contribute to understanding consumer ethics. Recent research has applied civic motivations and behaviors to various areas of marketing and consumption [10–13] but little research has explored a theoretical model of how civic engagement is related to various areas of marketing and consumption.

Second, little research has explored whether past experiences of unethical consumption (e.g., purchasing counterfeits) reduce the impact of ethical attitudes on consumption. Recent studies have illustrated that past unethical behaviors (e.g., buying or wearing counterfeit fashions) can lead to a greater possibility of cheating and other dishonest behaviors [14,15]. It can be hypothesized that past experience buying counterfeits can prompt further unethical consumption and reduce the impact of ethical attitudes on consumption behaviors.

Third, a nationally generalized sample was used to satisfy the principle that greater external validity results from a large representative sample. Many previous studies have relied on samples of college students or women to test ethical consumption attitudes and behaviors.

The purpose of this study was to examine how public service motivation (PSM) is related to ethical consumption behaviors and how past experience of unethical behavior can intervene between PSM and ethical consumer behaviors. No empirical research has explored the relationship between PSM and attitudes toward and the behavior of purchasing fashion counterfeits. It can be postulated that those with higher levels of PSM are less likely to purchase counterfeits and are more likely to disapprove of purchasing counterfeits. How does PSM influence purchasing of fashion counterfeits? How does PSM differ between those with and without past experience buying fashion counterfeits? More specifically, does PSM reduce the willingness to buy counterfeits, but does past experience buying counterfeits intervene between the positive impact of PSM and attitudes toward buying counterfeits?

## 2. Literature Review: Counterfeits, Consumer Ethics, Civic Virtue, and PSM

### 2.1. Purchasing Fashion Counterfeits and Ethics

Fashion products are among the most popular products to counterfeit. Fashion counterfeiting is a global issue. Despite various multinational and collaborative efforts, it is apparent that the problem of fashion counterfeiting is growing. Research examining various aspects of fashion counterfeits has been conducted around the world including America (e.g., U.S.), Europe (e.g., Italy), East Asia (e.g., China), and Middle East (e.g., Turkey). Some examples of the most recent studies include the following findings.

Park-Poaps and Kang [16], relying on U.S. female college students, examined the effects on purchase likelihood of: brand reputation (high vs. low), purchasing situation (counterfeit vs. genuine), product evaluation, and attitudes toward three fashion counterfeit products (shirt, handbags, shoes). Brand reputation impacted the likelihood of buying shirts and shoes. The purchasing situation affected purchase likelihood of the shirt. Attitudes toward fashion counterfeit products did not affect purchase likelihood, but product evaluations did affect likelihood of purchase across the three product types. Since Italy is especially affected by luxury product counterfeiting, Morra et al. [17] investigated the net impact of social media marketing (SMM), user-generated content (UGC), and firm-created content (FCC) on overall brand equity (OBE) and purchase intention toward genuine and counterfeit fashion luxury products. Italian undergraduate students were surveyed. Results showed that OBE and purchase intention toward fashion counterfeit products were positively influenced by UGC. Also, there was significant impact of OBE and FCC on purchase intention toward genuine luxury brands.

There has not been much research about Asian consumers' ethical values or attitudes toward counterfeiting, although Asian countries have the largest counterfeit market in the world [18]. For instance, more than 90% of products in the Chinese market are counterfeited, including music, movies, software, and fashion goods such as clothing, footwear, and accessories, consisting of about 57% of counterfeit products in the world [19]. Thus, Kozar and Huang [20] investigated the relationship among consumers' knowledge of counterfeits, face-saving, materialism, and ethical values with their attitudes toward fashion counterfeits by surveying more than 1000 participants in China. There was a significant relationship between Chinese consumer's knowledge of counterfeit goods, face-saving, materialism and attitude toward ethical values and counterfeiting. Gültekin [21] also explored the impact of the love of money (richness, importance, achievement, and budget) and ethical judgment on consumer intent to purchase fashion counterfeits. Consumers in Ankara, Turkey, who had purchased clothing counterfeits, participated in the study. The love of money (achievement and budget dimensions) had a positive impact on purchase intention toward counterfeit goods, while moral judgment about purchase of counterfeits was negatively related to purchase intention toward counterfeits. The rich and importance dimensions of love of money on counterfeit purchase intention was not significant.

These recent studies have only just scratched the surface of uncovering information that will serve to deter consumers' purchase of counterfeit fashion goods. Additional research is needed to uncover variables that can contribute to an understanding of counterfeit purchasing with ethical motivation and civic virtue.

## 2.2. Ethical Concerns about Consumption and Civic Virtue

Research on business and consumer ethics concentrating on both the supply and demand sides has examined diverse ethical issues from production of and traffic in illicit goods to the dark side of consumer behavior including the purchase of counterfeits and consumer fraud [5,7]. Despite publicity and attention directed to the supply side of illicit and unethical production, consumer misdemeanors have been increasing.

Research on business and consumer ethics has examined ethical issues from production of and trade in illegal goods to consumer purchase of counterfeits and consumer fraud [5–7,22,23]. Research on consumer ethics has addressed consumer activism [24,25], consumer boycotts [26,27], voluntary simplicity [28], ethical consumerism [29], responsible consumption [30,31], and consumer ethics in consumer misbehavior [3,32]. However, these studies have not addressed consumer ethics in terms of civic engagement motivation.

Good civic virtue can contribute to eliminating unethical issues in the marketplace. Civic virtue derived from a strong public service motivation emphasizes public interest and protection of the common good, which can control various consumer misbehaviors. Consumers who are good citizens hold corporations and governments to higher moral and quality standards [33]. Although 'citizen' and 'consumer' are considered separate categories and perhaps mutually exclusive domains [34,35],

PSM can transform consumerism for individual self-interest into civic virtue rooted in collective responsibility to a community commons. For instance, consumers with a high level of PSM emphasize the duty of citizens to buy goods from companies with a record of socially conscious production [36]. Consumers with high PSM are likely to manifest market virtues including integrity, honesty, and responsibility [37].

### 2.3. Neglected Area: Consumer Ethics and Public Service Motivation

Research on consumer ethics has explored moral values related to consumer decision-making. For instance, Brinkmann and Peattie (2008) [38] provide a theoretical link between moral intensity and purchase decisions in terms of private-hedonism motives (e.g., excitement and pleasure), private-caring motives (e.g., caring consumer behavior), private-social motives (e.g., consumption by social networks) and public motives (e.g., political participation and boycotts). However, research has confirmed an attitude-behavior discrepancy in ethical consumer behavior. The concept of neutralization explains how consumers justify their behavior as a means of coping with decision conflict and insulating themselves from blame and guilt. When consumers purchase counterfeits due to non-ethical factors (e.g., price, quality, availability, and brand reputation), their ethical beliefs can accommodate such consumption. For instance, consumers with strong PSM towards ethical purchase decisions would not be expected to buy counterfeits. Although people's ethical concerns are often not manifest in their behavior [30], there might be a strong link between public values and public service motivation that could transform ethical interests to actual ethical consumption. PSM includes a wide range of motives and actions in the public domain that are intended to do good for others and shape the well-being of society [39]. It can be hypothesized that PSM for civic virtue has a positive effect on ethical consumption and consumption morality. Future study can develop a rich research area for understanding a neglected relationship between civic virtue and purchasing counterfeits.

Three major trends in public service motivation research include: (1) the relationship between public service motivation and organization membership, (2) performance and innovation, and (3) ethical behavior. In particular, in the realm of ethical behavior, the effects of public service motivation on behavior are manifest in both social and organizational contexts. This general ethical orientation is associated with the willingness for self-sacrifice that accompanies public service motivation. The organizational effects of public service motivation are manifest in greater levels of behavior serving the public interest and in higher levels of organizational citizenship behaviors [40]. Recent studies [41,42] also report that the higher level of PSM, the higher level of innovativeness and creativity. High public service motivation is related to greater levels of altruism, behavior directly intended to help other people, and conscientiousness, the willingness to be indirectly helpful to others within the organization [43]. These civic components of PSM are expected to encourage development of civic assets, respect for creativity and originality and, subsequently, ethical consumption.

### 2.4. Moral Repercussions from Past Unethical Behaviors

It is expected that PSM will promote ethical consumption, that is, consumers with a high level of PSM will be less willing to buy counterfeits. However, the impact of PSM on ethical consumption may be moderated by previous experiences. For instance, past experience buying counterfeits may intervene between PSM and the purchase of counterfeits. Recent studies suggest that consumers with past experience buying counterfeits are more likely to buy them in the future [15,44]. That is, once having engaged in an ethical or unethical act, individuals are likely to behave the same way in the future. Gino, Norton, and Ariely [15] demonstrated that participants who wore counterfeit sunglasses were more likely to cheat, compared with participants who wore branded sunglasses. Moral disengagement allows people to repackage their beliefs regarding ethically questionable behavior so that the behavior is re-construed as ethically permissible [45,46]. When ethical boundaries are crossed, people are more likely to forget moral codes [44]. In sum, PSM is expected to play a strong role of reminding consumers of ethical codes. However, past experiences buying counterfeits can reduce

the intensity of PSM's impact. There might be significant differences in the impact of PSM on the willingness to buy counterfeits between those with and without past experience buying counterfeits.

## 3. Empirical Models and Data

### 3.1. Empirical Regression Models: The Impact of PSM

We apply PSM theory to consumer ethical behaviors. PSM refers to an "individual's predisposition to respond to motives grounded primarily or uniquely in public institutions and organizations" [47,48]. Perry and colleagues developed the theory of public service motivation in terms of three dimensions: rational, normative, and affective [48–51]. Rational motives relate to participation in public policy-making to pursue community interests and common goods for utility maximization. Norm-based motives compel people to commit to public interest, civic duty, and social justice. Affective motives allow people to come together and collaborate for the good of community bonding with the disadvantaged and vulnerable people. Based on the integration between three motivations and three public values, Perry [50] proposed six dimensions of PSM including attraction to public policy making, commitment to the public interest, self-sacrifice, compassion, civic duty and social justice. Researchers have also developed highly reliable instruments to measure dimensions of consumer ethics (e.g., Hunt-Vitell model or Muncy-Vitell Consumer Ethics scale; [3]). However, these scales do not directly measure why consumers buy counterfeits and the scales only illustrate a link between evaluation of various moral situations and the degree of willingness to buy counterfeits. Consumer ethical scales to directly measure ethical motivation will lead to a better understanding of what kind of ethical motivation factors are directly related to ethical consumption and why. Scales measuring motivation for ethical consumption can identify what ethical mechanisms lead to ethical consumption.

We argue that PSM theory can improve our understanding of citizens as consumers and the nature of ethical dimensions of consumer citizenship behavior. PSM may have a significant relationship with various types of ethical consumption behaviors. Research regarding a relationship between PSM and ethical consumption behaviors can disclose unexplored areas of consumer behavior and civic engagement.

Little empirical research has explored a relationship between PSM and consumer behavior. In particular, little research has explored how publicly related motives influence ethical consumer behaviors. For instance, Shaw et al. [52] suggests there are several publicly related motives including security (e.g., feeling that others care about me), benevolence (e.g., helpfulness and forgiveness), and universalism (e.g., equality, social justice, and tolerance). However, past studies has not empirically tested a relationship between these public motives and consumer decision making. While public policy research has developed the theory of public motivation, it has not been applied to consumer research, despite an increasing importance of the interaction between citizen and consumer within a globalized market. This study applies the theory of public motivation to a consumer behavior issue, that is, consumer ethics.

Three empirical models were used to test the impact of PSM on willingness to buy counterfeits. The estimations of Model 1-1 and Model 1-2 are based on an ordinary least square (OLS) regression. The estimation of Model 2 is derived from an ordered logit regression. After controlling for income, education, gender, and age, the following regression models were used.

The first test examined whether or not PSM is related to willingness to buy counterfeits (see Model 1-1). In Model 1-1, $\beta_1$ is expected to be a negative value and PSM is expected to have a negative impact on buying counterfeits. The hypothesis is that consumers with a higher (vs lower) level of PSM are not as willing to buy counterfeits. Also tested is whether past experience buying counterfeits increases the willingness to buy counterfeits. In Model 1-1, $\beta_{11}$ is expected to be a positive value. The size of $\beta_{11}$ is the difference in willingness to buy counterfeits between groups with and without experience buying counterfeits. Figure 1 shows all these hypotheses.

Willingness to buy counterfeits

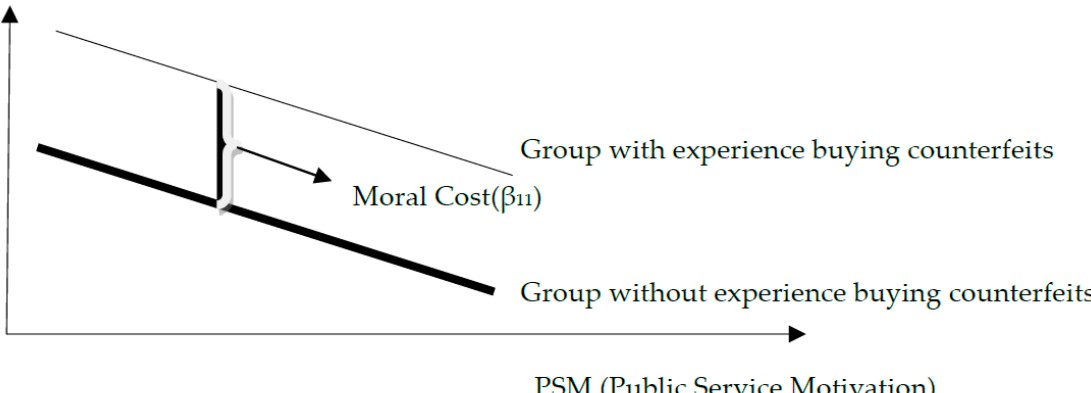

**Figure 1.** A relationship between PSM and unethical consumer behavior.

The second test explored whether past experience buying counterfeits would intervene between the negative impact of PSM and willingness to buy counterfeits (See Model 1-2). Figure 2 shows the negative impact of PSM on willingness to buy counterfeits is larger for those without experience than for those with experience. An interaction term (measured by the size of $\beta_{12}$) was used to test the relationship between PSM and the willingness to buy counterfeits. In Model 1-2, if $\beta_{12} > 0$, then the total negative impact of PSM is larger in those without past experience buying fashion counterfeits than in those with such experience.

Willingness to buy counterfeits

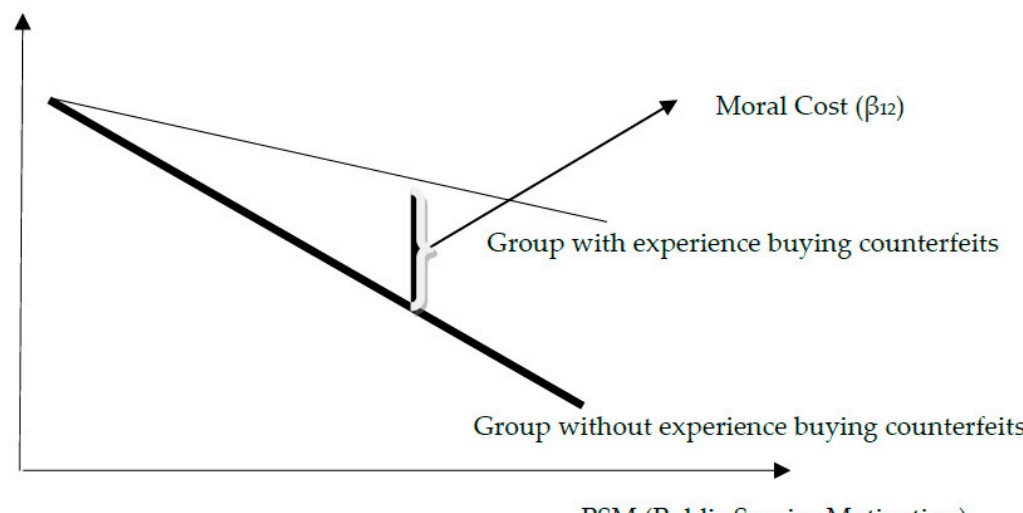

**Figure 2.** A relationship between PSM and unethical consumer behavior through past experience buying counterfeits.

The third model examined what specific elements of PSM were negatively related to the willingness to buy counterfeits (Model 2). Perry [50] hypothesized that attraction to public policy-making, commitment to public interest, social justice, civic duty, compassion, and self-sacrifice were six dimensions of PSM and suggested the four empirical components of the PSM construct including (1) commitment to the public interest and civic duty, (2) compassion, (3) attraction to public policy making, and (4) self-sacrifice. These dimensions are transformed into three motivational components in terms of public value (for what), altruistic attitude (for whom) and civic participation

behavior (how). We selected twelve items from Perry's measurement scale [50] in order to measure PSM and adopted three factors derived from factor analysis, including (1) public interest as citizen value, (2) compassion as citizen attitude, and (3) attraction to public policy making as citizen behavior (See Tables 1 and 2). The effects of these three factors of PSM represent $\beta_{21}$, $\beta_{22}$, and $\beta_{23}$ respectively in Model 2.

**Table 1.** Factor analysis of PSM.

| | Variables | Factor 1 | Factor 2 | Factor 3 | Uniqueness |
|---|---|---|---|---|---|
| | P4 | 0.7502 | 0.0370 | −0.0064 | 0.4357 |
| | P5 | 0.7290 | 0.2754 | 0.0639 | 0.3886 |
| | P6 | 0.7894 | 0.0690 | 0.0144 | 0.3719 |
| | P11 | 0.7634 | −0.0771 | −0.0359 | 0.4101 |
| Cronbach Alpah | P13 | 0.6206 | 0.3539 | −0.0003 | 0.4895 |
| of 12 PSM items | P14 | 0.7440 | 0.1461 | 0.0001 | 0.4251 |
| scale reliability | P7 | 0.1649 | 0.8100 | −0.0451 | 0.3146 |
| coefficient: | P8 | 0.0279 | 0.8240 | −0.0272 | 0.3196 |
| 0.769 | P9 | −0.0588 | 0.6234 | 0.3537 | 0.4828 |
| | P12 | 0.4369 | 0.6189 | −0.0095 | 0.4260 |
| | P1 | 0.0288 | −0.1375 | 0.8352 | 0.2828 |
| | P2 | −0.0069 | 0.1478 | 0.8330 | 0.2843 |

| Factor | Variance | Proportion | Cumulative |
|---|---|---|---|
| Factor 1 | 3.462 | 0.289 | 0.288 |
| Factor 2 | 2.382 | 0.199 | 0.487 |
| Factor 3 | 1.525 | 0.127 | 0.614 |

Notes: (1) Factor Analysis Method: principal-component factors from STATA 13; Retained factors = 3; Rotation: Orthogonal Varimax method (Kaiser off); Number of parameters = 33. (2) N = 3188.

**Table 2.** Questions for factor analysis of PSM.

| Factors | Questions |
|---|---|
| Factor 1<br>(F1: Public interest) | 5-point Likert scale: strongly disagree (1) to strongly agree (5)<br>P4. I contribute to community interest beyond self-interest.<br>P5. It is important for me to contribute to public interest.<br>P6. I do my best for the whole community even if it harmed my interests.<br>P13. I think people should give back to society more than they get from it.<br>P11. I usually do a lot of work for public interest than myself.<br>P14. I am prepared to make enormous sacrifices for the good of society. |
| Factor 2<br>(F2: Compassion) | 5-point Likert scale: strongly disagree (1) to strongly agree (5)<br>P7. It is difficult for me to contain my feelings when I see people in distress.<br>P8. Most social programs are too vital to do without.<br>P9. I consider the welfare of people I don't know personally.<br>P12. Serving other citizens would give me a good feeling even if no one paid me for it. |
| Factor 3<br>(F3: Attraction to Public Policy Making) | 5-point Likert scale: strongly disagree (1) to strongly agree (5) (Reverse coding)<br>P1. I have a negative view on political negotiation.<br>P2. I do not like negotiation and coordination involved in public policy process. |

Model 1-1

$$Yc_i = \alpha_{11} + \beta_1 PSM_i + \beta_{11}Exp\_Buy_i + \beta_3 Female_i + \beta_4 Single_i + \beta_5 College_i + \beta_{61}Income_i + \beta_{62}Income^2_i + \beta_{63}Income^3_i + \beta_{71}Age_i + \beta_{72}Age^2_i + \beta_{73}Age^3_i + \beta_8 Metro_i + \varepsilon_{11i}$$

Model 1-2

$$Yc_i = \alpha_{12} + \beta_1 PSM_i + \beta_{12}(PSM_i * Exp\_Buy_i) + \beta_3 Female_i + \beta_4 Single_i + \beta_5 College_i + \beta_{61}Income_i + \beta_{62}Income^2_i + \beta_{63}Income^3_i + \beta_{71}Age_i + \beta_{72}Age^2_i + \beta_{73}Age^3_i + \beta_8 Metro_i + \varepsilon_{12i}$$

Model 2

$$Yo_{ij} = \alpha_2 + \beta_{21}F1_{ij} + \beta_{22}F2_{ij} + \beta_{23}F3_{ij} + \beta_{24}(F3xF3)_{ij} + \beta_3Female_{ij} + \beta_4Single_{ij} + \beta_5College_{ij} + \beta_{61}Income_{ij} + \beta_{62}Income^2_{ij} + \beta_{63}Income^3_{ij} + \beta_{71}Age_{ij} + \beta_{72}Age^2_{ij} + \beta_{73}Age^3_{ij} + \beta_8Metro_{ij} + \varepsilon_{2ij}$$

$i$ = respondents (1,2,3, . . . N); $j$ = experience buying counterfeits (1 = Yes, 2 = No)

Yc = Willingness to buy fashion counterfeits (A scale from 10 to 90)

Yo = Willingness to buy fashion counterfeits (A five point Likert scale)

PSM = Public Service Motivation (Sum of 12 question items)

F1 = Factor variable representing one sub-element of PSM (Public Interest)

F2 = Factor variable representing one sub-element of PSM (Compassion)

F3 = Factor variable representing one sub-element of PSM (Attraction to Public Policy Making)

F3 x F3 = The square variable of F3

Exp_Buy = Past experience buying fashion counterfeits (Yes = 1, No = 0)

$PSM_i$*Exp_Buy = Interaction term between PSM and Exp_Buy

Female = Dummy variable (If respondents are female, Female = 1; otherwise Female = 0)

Single = Dummy variable (If respondents are single, Single = 1; otherwise Single = 0)

College = Dummy variable (If respondents have college degree, College = 1; otherwise College = 0)

Income = The level of income (Monthly household income from 0 to 11)

Age = Respondents' age

Metro = Dummy variable (If respondents are living in metropolitan areas, Metro = 1; else Metro = 0)

### 3.2. Data and Measurements

A web-based on-line survey (Gallup Korea) was conducted with South Korean citizens 19 years old or older from January 15th to 30th, 2013. Of the 5000 sampled, 3189 respondents completed the survey for a response rate of 63.78%. The sample is nationally representative in terms of gender (male = 51.2%; female = 48.8%), age (range = 19 to 81; mean age = 43.41), and regions (including seven metropolitan cities and nine provinces). The questionnaire included demographic items, items about public service motivation, and items about purchase of fashion counterfeit products (e.g., How likely do you agree that it's okay to purchase counterfeit fashion products? 5-point Likert scale strongly agree to strongly disagree. Have you ever purchased counterfeit fashion products? Yes/No) [53] (see Table A1 for variables and measurements).

The elements of PSM appear to be an integrative concept to capture the nature of ethical attitudes and intentions for ethical consumption or against unethical consumption. The PSM measurements include cognitive and affective moral elements. In addition, some PSM elements contain interest in and readiness for public interest. The concept of PSM is comprehensive enough to capture or explain the mechanism between ethical attitudes and ethical consumption. Example items from Perry's (1990) PSM scale [48] are: public interest (e.g., I think people should give back to society more than they get from it), compassion (e.g., It is difficult for me to contain my feelings when I see people in distress), and attraction to public policy making (e.g., I have a negative view on political negotiation). Data analysis included descriptive statistics, factor analysis, and regression analyses (see Table A2 for simple statistics).

## 4. Empirical Findings and Discussion

First, empirical findings from both Model 1-1 and Model 1-2 show a negative impact of PSM on the willingness to buy fashion counterfeits. Table 1 illustrates a strong negative impact of PSM on

the willingness to purchase counterfeits in Model 1-1, which is statistically significant (Estimate of PSM = −0.325; Standard error of PSM = 0.055, *p*-value < 0.01).

Second, there was a significant negative interaction between PSM and past experience buying counterfeits in Model 1-2. An estimate of the interaction term is +0.238 with 0.120 standard error (*p*-value < 0.05), which means past experience buying counterfeits reduces the negative impact of PSM on the willingness to buy counterfeits (see Model 1-2 of Table 3). The statistically significant effect of the interaction term suggests that PSM can fully contain a causal mechanism, even though common method bias can be present between PSM and the willingness to buy counterfeits. For consumers with past experience buying counterfeits, the size of the negative impact of PSM is reduced to −0.167 (= −0.405 + 0.238). These results imply that past experience buying counterfeits reduces the impact of PSM on ethical consumer behaviors. In other words, the negative impact of PSM is stronger (approximately −0.405) for those without past experience buying counterfeits reinforcing their unwillingness to purchase counterfeits.

**Table 3.** Regression results: purchasing counterfeits and PSM.

| Dependent Variable (The Willingness to Buy Fashion Counterfeits) | Model 1-1 Parameter Estimate | Model 1-2 Parameter Estimate |
|---|---|---|
| Intercept | 80.301 *** (8.687) | 83.809 *** (8.832) |
| PSM | −0.325 *** (0.055) | −0.405 *** (0.066) |
| Exp_Buy | 7.123 *** (0.583) | −2.340 (4.416) |
| PSM*Exp_Buy | | 0.238 ** (0.120) |
| Female | 0.933 * (0.573) | 0.930 (0.573) |
| Single | 0.170 (0.901) | 0.134 (0.901) |
| College | −0.452 (0.613) | −0.440 (0.613) |
| Income | 1.960 * (1.086) | 1.969 * (1.086) |
| Income$^2$ | −0.439 ** (0.215) | −0.439 ** (0.215) |
| Income$^3$ | 0.027 ** (0.012) | 0.027 *** (0.012) |
| Age | −2.259 *** (0.587) | −2.279 *** (0.587) |
| Age$^2$ | 0.055 *** (0.013) | 0.055 *** (0.013) |
| Age$^3$ | −0.00042 *** (0.000099) | −0.00042 *** (0.000098) |
| Metro | 1.241 ** (0.556) | 1.237 ** (0.556) |
| R-square | 0.0699 | 0.0713 |
| F-value | 19.88 *** (*df* = 12, 3175) | 18.73 *** (*df* = 13, 3174) |

Notes: (1) Numbers in parentheses are standard errors. (2) ***, **, * indicates statistical significance at the 1%, 5%, and 10% level, respectively. (3) N = 3188.

Third, more specifically, which of the three underlying factors of PSM, that is, public interest, compassion, and attraction to public policy making are more or less associated with the willingness to buy counterfeits? (see Table 4). The results of the whole sample including those with and without past experience of purchasing fashion counterfeits show that all three elements of PSM (public interest,

compassion, and attraction to public policy making) are significantly negatively related to willingness to purchase fashion counterfeits (see Table 4). In addition, the impact of the attraction to public policy making (F3) on purchasing counterfeits is concave, where its impact is not linear with the negative quadratic terms. The linear impact ($\beta_{23}$) is not statistically significant but the quadratic impact ($\beta_{24}$) is negatively significant, implying that the negative marginal impact of the attraction to public policy making is increasingly large. Furthermore, it should be noted that there is a significant difference in the impact of PSM on the willingness to buy counterfeits between groups with and without past experience to buy counterfeits. The negative effects of PSM are statistically significant for those without past experience of purchasing fashion counterfeits (public interest, $p < 0.05$; compassion, $p < 0.01$; attraction to public policy making, $p < 0.05$) (see Table 4). All three factors of PSM are not significant for those who had purchased counterfeits. In addition, women (compared with men), younger (compared with older) respondents, and respondents living in metropolitan areas (compared with non-metropolitan areas) are more likely to agree with purchasing fashion counterfeits (See Table 4).

**Table 4.** Ordered logit regressions (Model 2): the impact of PSM on the willingness to buy counterfeits.

| | Whole Group (N = 3188) | | Group without Experience to Buy Fashion Counterfeits (N = 2054) | | Group with Experience to Buy Fashion Counterfeits (N = 1134) | |
|---|---|---|---|---|---|---|
| | Parameter Estimate | | Parameter Estimate | | Parameter Estimate | |
| F1 (Public Interest) | −0.096 ** | (0.038) | −0.094 ** | (0.047) | −0.087 | (0.065) |
| F2 (Compassion) | −0.210 *** | (0.036) | −0.263 *** | (0.044) | −0.088 | (0.064) |
| F3 (Attraction to Public Policy Making) | −0.041 | (0.035) | −0.058 | (0.043) | −0.017 | (0.062) |
| F3 Square (= F3 × F3) | −0.065 ** | (0.024) | −0.087 ** | (0.029) | −0.015 | (0.042) |
| Exp_Buy | 0.916 *** | (0.076) | - | - | - | - |
| Female | 0.131 * | (0.072) | 0.155 * | (0.089) | 0.081 | (0.123) |
| Single | 0.059 | (0.112) | 0.042 | (0.140) | 0.086 | (0.187) |
| College | −0.087 | (0.076) | −0.215 ** | (0.094) | 0.178 | (0.132) |
| Income | 0.257 * | (0.137) | 0.133 | (0.163) | 0.514 ** | (0.254) |
| Income$^2$ | −0.058 ** | (0.027) | −0.024 | (0.033) | −0.122 ** | (0.049) |
| Income$^3$ | 0.004 ** | (0.002) | 0.0015 | (0.0019) | 0.007 ** | (0.003) |
| Age | −0.282 ** | (0.073) | −0.149 * | (0.090) | −0.590 *** | (0.127) |
| Age$^2$ | 0.007 ** | (0.0017) | 0.0039 * | (0.002) | 0.014 *** | (0.0029) |
| Age$^3$ | −0.00005 ** | (0.00001) | −0.000033 ** | (0.000015) | −0.000098 *** | (0.00002) |
| Metro | 0.140 *** | (0.069) | 0.014 | (0.085) | 0.416 *** | (0.121) |
| Intercept$_1$ | −5.370 | (1.037) | −3.742 | (1.264) | −10.352 | (1.835) |
| Intercept$_2$ | −3.711 | (1.034) | −2.140 | (1.262) | −8.414 | (1.826) |
| Intercept$_3$ | −0.734 | (1.033) | 0.863 | (1.262) | −5.423 | (1.814) |
| Intercept$_4$ | 2.034 | (1.051) | 3.540 | (1.295) | −2.563 | (1.832) |
| | *LR chi2(14)* = 250.47 | | *LR chi2(14)* = 99 | | *LR chi2(14)* = 48.22 | |
| | *Prob > chi2* = 0.00001 | | *Prob > chi2* = 0.00001 | | *Prob > chi2* = 0.00001 | |
| | *Log likelihood* = −3570.88 | | *Log likelihood* = −2352.29 | | *Log likelihood* = −1197.47 | |

Notes: (1) Numbers in parentheses are standard errors. (2) ***, **, and * indicates statistical significance at the 1%, 5%, and 10% level, respectively.

## 5. Conclusions and Future Research

Respondents with a higher level of PSM were less willing to buy counterfeits. Respondents with past experience buying counterfeits were more willing to buy counterfeits than respondents without such experience. This finding suggests that consumer morality can matter to buying counterfeits [5,10,54].

Findings illustrated a negative interaction between PSM and past experience buying counterfeits. Past experience buying counterfeits intervened between the impact of PSM and willingness to buy counterfeits such that the impact of PSM was weakened. Past experience buying counterfeits appears to involve moral compromises that fail to constrain the willingness to buy counterfeits. Recent studies have reported moral repercussions derived from past unethical experiences [14,15]. Our findings suggest that past experience to buy counterfeits can lead to a practice of buying counterfeits through

weakening a moral mechanism because the past moral disengagement like purchasing counterfeits makes consumers more likely to be involved in further misdemeanors.

Previous research has focused on why consumers purchase counterfeit products [45,46,53,54], but has not provided practical implications from a public policy perspective. The empirical results of this study suggest that prior purchase of counterfeits can constrain or eliminate a strong negative effect of public service motivation on willingness to purchase counterfeits. Past experience buying counterfeits should be further examined in terms of gender, age and education. Purchase of fashion counterfeits is widespread and global, especially in Asian countries such as China, Japan, and Korea. This consumption has a negative effect on legitimate fashion industries. Thus, a new social consensus against purchasing counterfeits needs to be nurtured to discourage counterfeit purchases. For example, public policy campaigns with a goal to inform the public about the negative impact of purchasing counterfeit products may contribute to reducing the purchase of counterfeits. Several such campaigns have proven to be successful strategies in changing ethical attitudes and behaviors.

Our empirical findings supporting a strong relationship between public service motivation and ethical consumer behaviors lead to theoretical and practical implications in public policy and marketing. The current analysis of civic engagement motivation in consumer behaviors derived from public service motivation provides a new perspective on a rational choice model between motivation and behavioral change. Beyond the rational choice model emphasizing economic incentives, further research is required about how non-economic motivations can promote ethical behavioral changes in marketing and business research. Public- and community-based values inherent in public service motivation can provide a starting point for a search for innovative ways to change consumer behaviors. Cross-cultural studies comparing public service motivation and consumer behavior may provide useful strategies to boost ethical consumption.

Our findings are also expected to facilitate future research on potential relationships among public values, business ecosystems, and open innovation [55–57]. Public service motivation can play a mediating role of connecting social responsibility and entrepreneurial business activities; thus, generating a constructive relationship between long-term orientation necessary for sustainability and responsible innovation. Further research will be focused on how public service motivation can strengthen business ethics and open innovation within a cohesive entrepreneurial ecosystem and long-term sustainability [58].

Our empirical results have limitations to identify a causal mechanism of PSM on ethical consumption. Researchers should be mindful of the likelihood of common method bias even though we find both the significant effects of the three distinct factors of PSM through factor analysis and the significant interaction effect between PSM and past experience buying counterfeits. The negative impact of PSM on purchasing counterfeits may be overestimated if some respondents tend to overemphasize both PSM and ethical consumption behaviors because of social desirability bias. While the common method bias cannot inflate quadratic and interaction effects of PSM [59,60], it is still necessary to control for a potential unknown bias for a relationship between PSM and consumer misbehaviors. In order to reduce the method bias, future research can use multiple data sources for dependent and independent variables and procedural remedies of survey process as well as statistical ones. Experimental approach with separate measurements of PSM and ethical consumption behaviors is highly recommended.

**Author Contributions:** K.J., S.-H.L. and J.W. conceived and designed the research idea and analytical frame. K.J. organized the survey in South Korea and analyzed the data; K.J., S.-H.L. and J.W. wrote the paper.

**Funding:** This work was supported by the National Research Foundation of Korea Grant funded by the Korean Government (NRF-2017S1A3A2066084).

**Acknowledgments:** This paper is based on the 2013 International Textile and Apparel Association (ITAA) conference presentation with the title, "Public Service Motivation (PSM) and Attitudes Toward Purchasing Fashion Counterfeits" and revised substantially in terms of new literature review and research questions, empirical models and statistical methods, and new discussions with various policy implications and research agenda.

**Conflicts of Interest:** No potential conflict of interest was reported by the authors.

**Appendix A**

<div align="center">

**Table A1.** Variables and Measurements.

</div>

| Variable | Definitions and Measurements |
| --- | --- |
| Yc | Generally how likely do you agree that it's okay to purchase counterfeit products? ① Strongly agree to purchase ② Agree ③ Neutral ④ Disagree ⑤ Strongly disagree |
| Yo | Recoding Yc: ① to 10; ② to 30; ③ to 50 ④ to 70; and ⑤ to 90 |
| PSM | Sum of score the following twelve items about public service motivation: P1, P2, P4, P5, P6, P7, P8, P9, P11, P12, P13, P14 |
| Exp_buy | Experience to purchase counterfeits (If respondents have experiences to buy fashion counterfeits, Exp_buy = 1; otherwise Exp_buy = 0) |
| PSM_EB | Interaction term between PSM and Exp_buy |
| F1 | Factor score representing 'public interest' (Standard normal distribution with a mean of '0' and a standard deviation of '1') |
| F2 | Factor score representing 'compassion' (Standard normal distribution with a mean of '0' and a standard deviation of '1') |
| F3 | Factor score representing 'attraction to public decision making' (Standard normal distribution with a mean of '0' and a standard deviation of '1') |
| Female | If respondents are female, then Female = 1; otherwise Female = 0 |
| Single | If respondents are single, then Single = 1; otherwise Single = 0 |
| College | If respondents have four-year college degrees, College = 1; otherwise College = 0 |
| Income | Respondents' household monthly income from 0 to 11; 0 = No income, 1 = 5, 2 = 15, 3 = 25, 4 = 35, 5 = 45, 6 = 55, 7 = 65, 8 = 75, 9 = 85, 10 = 95, 11 = 100 (Unit = one hundred and thousand Korean Won) |
| $Income^2$ | Income*Income |
| $Income^3$ | Income*Income*Income |
| Age | Respondents' age |
| $Age^2$ | Age*Age |
| $Age^3$ | Age*Age*Age |
| Metro | If respondents live in metropolitan areas, Metro=1; otherwise Metro=0 |
| P1 to P14 | Measurements for PSM. For more details see Table 2. |

<div align="center">

**Table A2.** Simple Statistics (N = 3188).

</div>

| Variable | Mean | Standard Deviation | Minimum | Maximum |
| --- | --- | --- | --- | --- |
| Yc | 2.666 | 0.807 | 1 | 5 |
| Yo | 43.325 | 16.145 | 10 | 90 |
| PSM | 39.609 | 5.309 | 12 | 60 |
| Exp_buy | 0.356 | 0.479 | 0 | 1 |
| PSM_EB | 14.156 | 19.302 | 0 | 60 |
| F1 | 0 | 1 | −3.632 | 3.353 |
| F2 | 0 | 1 | −4.266 | 2.860 |
| F3 | 0 | 1 | −3.511 | 3.579 |
| Female | 0.488 | 0.500 | 0 | 1 |
| Single | 0.318 | 0.466 | 0 | 1 |
| College | 0.571 | 0.495 | 0 | 1 |
| Income | 4.719 | 2.306 | 0 | 11 |

**Table A2.** *Cont.*

| Variable | Mean | Standard Deviation | Minimum | Maximum |
| --- | --- | --- | --- | --- |
| Income$^2$ | 27.582 | 26.468 | 0 | 121 |
| Income$^3$ | 188.718 | 278.652 | 0 | 1331 |
| Age | 43.401 | 13.628 | 19 | 81 |
| Age$^2$ | 2069.34 | 1214.803 | 361 | 6561 |
| Age$^3$ | 106,159.3 | 88,277.5 | 6859 | 531,441 |
| Metro | 0.505 | 0.500 | 0 | 1 |
| P1 | 2.810 | 0.824 | 1 | 5 |
| P2 | 3.185 | 0.832 | 1 | 5 |
| P4 | 2.973 | 0.856 | 1 | 5 |
| P5 | 3.258 | 0.785 | 1 | 5 |
| P6 | 3.012 | 0.868 | 1 | 5 |
| P7 | 3.916 | 0.769 | 1 | 5 |
| P8 | 4.029 | 0.804 | 1 | 5 |
| P9 | 3.437 | 0.879 | 1 | 5 |
| P11 | 2.884 | 0.783 | 1 | 5 |
| P12 | 3.668 | 0.761 | 1 | 5 |
| P13 | 3.347 | 0.802 | 1 | 5 |
| P14 | 3.089 | 0.840 | 1 | 5 |

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
