# Peer review of "Purchasing Counterfeits and Citizenship: Public Service Motivation Matters"

_sustainability, doi:10.3390/su11010103_

Round 1
Reviewer 1 Report
The paper is generally well presented. It identifies a gap within its field and provides a very sizable dataset to explore its hypothesis. In terms of contribution, the paper has found exactly what would have been expected, and is consistent with existing knowledge on both citizenship and the importance of past-experience in fashion purchasing. As such the paper is not providing a major or particularly significant contribution to the field, although is a good confirmatory study in a new context.
The structuring of the paper made it a little difficult to get into. The linkage between Citizenship and PSM is not made clear – and presented as an assumed reality that PSM is a measure of citizenship. This could have been clarified in section 2.1 with a couple of extra sentences evidencing this assumption.
The theoretical models are also conflated with the data presentation, which again makes readability that bit more difficult. Similarly the amalgamation of both results and methodology hinder readability. Making a clear argument for why the different modes of analysis were applied in the methods and a clearly separated results section would have aided readability.
The paper is also limited by its in-built assumptions regarding: responder bias (was any control for SDB implemented?), common method variance (someone self-identifying as high PSM would be less likely to identify buying counterfeit goods) and the abstraction of the method from a lived reality (fashion is a social/emotional consumption activity – not rational cognitive choice). A limitations section clearly outlining the limitations of this approach should therefore have been included. As the study also focused on a very limited number of independent variables, the extent to which the authors can say PSM or past-experience are good antecedents to counterfeit consumption prediction are also limited.
Overall though this is a good study of its type.
Author Response
We appreciate your constructive criticism. We tried to consider your key comments and revised the manuscript with substantial modification to improve the quality of the manuscript. More specifically, we deliberated your comments for the revision and tried to provide the following relevant discussions.
1. Clarifying the linkage between PSM and citizenship. We added several points to support this linkage
2. Making empirical models readable. We added several words to the models and made them easily understandable
3. Considering common method bias to identify the impact of PSM. We discussed the potential bias of the method bias and addressed the necessity of future research to minimize the bias. Social desirability can be considered as a source of omitted variable bias. We discussed this problem as limitation for our study with future research with experimental design with multiple sources of measurements of PSM and consumer misbehaviors. Two references are added in order to address this problem.
Again we greatly appreciate your comments on this manuscript.

Reviewer 2 Report
Manuscript ID: Sustainability-382559
Manuscript Title: Purchasing Counterfeits and Citizenship: Public Service Motivation Matters
I appreciate the opportunity to read this manuscript. It is good to see that the concept of public service motivation (PSM) is making its way into fields outside of Public Administration. That said, the manuscript has some very serious shortcomings that makes it hard for me to see this as a study of PSM. I am sorry I cannot leave a more positive review.
First, I cannot wrap my head around the use of PSM in the study’s context. PSM originated as a concept to study public sector employees’ motivation to deliver public services based on non-economic motives. Public services are services ordered and or paid for by government. Purchasing consumer goods is an entirely private matter. Does purchasing counterfeits have negative externalities for society at large – yes! But, to me, is does not make sense to posit PSM as a direct motivational component behind such behavior. Highly public service motivated individuals, perhaps, are likely to also be infused with strong values and ethics oriented towards collectivism and therefore would incur psychological costs from violating social norms (i.e., buying illegal or counterfeited goods). However, PSM is not the driver then, such values are.
Second, and almost as important, I am very concerned that the authors’ results are simply artefacts of their methodological approach. PSM is well-known to be prone to social desirability. In other words, it is socially accepted “to be committed to the public interest” etc. For your dependent variable, you ask items such “how likely do you agree that it is okay to purchase counterfeit fashion products?” (p. 7, lines 212-13). Clearly, it is socially desirable to offer negative answers to this question. The problem thus emerges as it is highly socially desirable to provide high scores on PSM items and low scores on counterfeit purchasing items, implying that your results can very easily be a product of individual differences in latent susceptibility to conform to social norms. In other words, endogeneity – omitted variable bias – is a very big problem for your study, and I am not at all convinced that your main results are picking up anything other than common source bias.
Author Response
We appreciate your constructive criticism. We tried to consider your key comments and revised the manuscript with substantial modification to improve the quality of the manuscript. More specifically, we deliberated your comments for the revision and tried to provide the following relevant discussions
1. Is PSM a driver to support ethical values? We added more details on the three characteristics of PSM including citizen values, citizen participation, and citizen attitude. In this sense, PSM is assumed as a driver to constrain the intention to buy counterfeits.
2. Considering common method bias to identify the impact of PSM. We discussed the potential bias of the method bias and addressed the necessity of future research to minimize the bias. Social desirability can be considered as a source of omitted variable bias. We discussed this problem as limitation for our study with future research with experimental design with multiple sources of measurements of PSM and consumer misbehaviors. Two references are added in order to address this problem.
Again we greatly appreciate your comments on this manuscript.

Reviewer 3 Report
I appreciate the opportunity to have read this paper. This research is tackling a gap in the literature and its topic of ethical consumption behaviors has been gaining increased attention in recent years. However, there are concerns that may limit the contribution of the investigation.
lines 45-47: “Recent ethical scales include items measuring...” Need citation for this statement.
Literature Review: This study incorporates PSM as a crucial factor affecting consumers’ counterfeit purchases, however, Literature Review provides insufficient information about this construct. Previous findings of the effect of PSM on consumer behaviors are needed. If that is hard to find, at least address its influence on performing other ethical behaviors.
There is a number of studies examining fashion counterfeit purchase behaviors. Please provide a summary of the findings in Lit Review.
Lines 141- 145: Authors wrote “Perry and colleagues developed the theory of public service motivation in terms of three dimensions: rational, normative, and affective…” They mentioned the first two dimensions but skipped the affective dimension- please address what this dimension involves.
Lines 191: What were the criteria for selecting 12 items from Perry’s full measurement of PSM? Authors continuously mention there are 6 dimensions in Perry’s scale but only 3 were tested.
p. 7 It would be more informative to compare South Korea’s population and sample in terms of education and income level to see whether the study’s sample is “a nationally representative sample from South Korea”.
Lines 240-243: “The results of the whole sample including those with and without past experience of purchasing fashion counterfeits show that all three elements of PSM (public interest, compassion, and attraction to public policy making) are significantly negatively related to willingness to purchase fashion counterfeits (see Table 3).” In Table 3, it shows that F3 (attraction to public policy making) is insignificant in all three columns. Please clarify.
Table 3: What does F3 x F3 mean?
Conclusions and Future Research: Although previous studies may have not used the construct PSM, a large number of studies of counterfeit purchase have examined conceptually somewhat similar constructs (e.g., ethical judgments, moral awareness, altruism) as factors influencing purchase behaviors. In addition, there are studies that examined the influence of previous purchase experience in this domain. Please compare the findings of this present study with previous findings.
Author Response
We appreciate your constructive criticism. We tried to consider your key comments and revised the manuscript with substantial modification to improve the quality of the manuscript. More specifically, we deliberated your comments for the revision and tried to provide the following relevant discussions
1. We included citations about scales on ethical consumption
2. Clarifying the linkage between PSM and citizenship. We added several points to support this linkage
3. Providing a literature review on the reason for why people buy fashion counterfeits. We added a page about the review. We also added more references.
4. Missing affective dimension of PSM. We added the dimension.
5. How many Perry’s dimensions of PSM are used? We clarified three dimensions of the six ones in Perry’s original version.
6. F3 x F3? We clarified this as a square term of the variable of F3.
7. Ambiguous interpretation of the impact of F3. We clarified the impact of F3 as quadratic form with some discussions of the nonlinear effect.
Again we greatly appreciate your comments on this manuscript.
Round 2
Reviewer 2 Report
I appreciate the authors' efforts in highlighting the limitations of their design. That said, they still pose a significant threat to the validity of results presented here. I cannot see how the authors will be able to increase our confidence in the associations presented without additional data collection that allows for a clearer identification strategy.
Reviewer 3 Report
All comments have been addressed and the manuscript has been improved after the revision.